# Engineered mRNA and the Rise of Next-Generation Antibodies

**DOI:** 10.3390/antib10040037

**Published:** 2021-09-26

**Authors:** Laura Sanz, Luis Álvarez-Vallina

**Affiliations:** 1Molecular Immunology Unit, Hospital Universitario Puerta de Hierro Majadahonda, 28220 Madrid, Spain; 2Cancer Immunotherapy Unit (UNICA), Department of Immunology, Hospital Universitario12 de Octubre, 28041 Madrid, Spain; 3Immuno-Oncology and Immunotherapy Group, Instituto de Investigación Sanitaria 12 de Octubre (imas12), 28041 Madrid, Spain

**Keywords:** RNA, engineered mRNA, antibody fragments, multispecific antibodies, next-generation antibodies

## Abstract

Monoclonal antibodies are widely used as therapeutic agents in medicine. However, clinical-grade proteins require sophisticated technologies and are extremely expensive to produce, resulting in long lead times and high costs. The use of gene transfer methods for in vivo secretion of therapeutic antibodies could circumvent problems related to large-scale production and purification and offer additional benefits by achieving sustained concentrations of therapeutic antibodies, which is particularly relevant to short-lived antibody fragments and next-generation, Fc-free, multispecific antibodies. In recent years, the use of engineered mRNA-based gene delivery has significantly increased in different therapeutic areas because of the advantages it possesses over traditional gene delivery platforms. The application of synthetic mRNA will allow for the avoidance of manufacturing problems associated with recombinant proteins and could be instrumental in consolidating regulatory approvals for next-generation therapeutic antibodies.

## 1. Introduction

In 2005, Philipp Holliger and Peter J. Hudson published a seminal review entitled “Engineered antibody fragments and the rise of single domains” [1] where the authors foresaw “a burgeoning range of regulatory approvals for recombinant monoclonal antibody (mAb) fragments in diagnosis and therapy”. There was a range of applications in which the Fc (fragment crystallizable)-mediated effects were not required. Recombinant antibody fragments, such as Fab (antigen-binding fragment), scFv (single-chain variable fragment), and single-domains (V_HH_ antibodies, also known as nanobodies), offered unprecedented advantages, such as increased tumor penetration [2] and/or binding to epitopes inaccessible for the large and flat antigen-binding sites of conventional antibodies [3,4]. 

Fifteen years later, this prediction has not been fulfilled. If we take a look at the antibody-based therapeutics approved or in revision devoid of the Fc region, they are still a minority. Only 10 out of 120 are not based on canonical IgG formats as of September 2021 (Table 1). This short list includes four Fabs; one scFv; two scFv-based immunotoxins; a bispecific tandem (scFv)_2_, also known as ‘bispecific T cell engager’ (BiTE); one monospecific bivalent nanobody, and a soluble T cell receptor (TCR) fused to a scFv. Removal of the Fc domain is a standard approach to prevent adverse immune effects, but with the caveat of abolishing Fc-driven cellular recycling via neonatal Fc receptor (FcRn) engagement, which leads to a significant reduction in blood residence time [1,5]. Alternatively, antibody fragments can be introduced into enclosed locations where rapid renal clearance is not an issue. This is the case for vascular endothelial growth factor (VEGF) blockers ranibizumab (Fab) and brolucizumab (scFv), which are used for the treatment of age-related macular degeneration and administered by intravitreal injection. The scenario in late-stage antibodies is no better: 5 out of 99 (Table 2), practically half the percentage. Even non-conventional formats such as bispecific antibodies mostly retain their Fc regions, and platforms that were initially designed Fc free have included silenced Fc domains in their most recent designs in an attempt to extend their circulatory half-life, such as the HLE (half-life extended) BiTE and [6] Fc-DART (Dual-Affinity Re-Targeting) antibodies [7]. 

## 2. The Challenges of Antibody Fragment Development

What was wrong, then, with Holliger and Hudson’s prediction? Was the incorporation of Fc domains the only option to increase the half-life of antibodies? Or was industry reluctant to change its manufacturing procedures? Several studies have shown that it is possible to extend the half-life of antibody fragments through fusion with albumin-binding peptides [8] or proteins [9,10,11]. Half-life extension of nanobodies via fusion to an albumin-binding nanobody has been demonstrated in different species in preclinical studies [12]. In addition, vobarilizumab (formerly ALX-0061) reached a phase 2b clinical trial in patients with moderate to severe rheumatoid arthritis, demonstrating a positive impact on disease activity [13]. Vobarilizumab is a bispecific nanobody with a high affinity for the IL-6 receptor, combined with an extended half-life by targeting human serum albumin (HSA). An apparent plasma half-life of 6.6 days was observed after a single intravenous administration of 10 mg/kg ALX-0061 in non-human primates, similar to the estimated expected half-life of HSA [14]. Ozoralizumab (ATN-103, TS-152), a trivalent, bi-specific nanobody that potently neutralizes tumor necrosis factor alpha (TNFα) and binds to HSA to increase its in vivo half-life is in late-stage clinical trials (phase 3) [15] and has been submitted for approval in Japan (2021). An anti-hepatocyte growth factor nanobody has also been molecularly fused to an albumin-binding nanobody unit to obtain serum half-life extension [16].

In addition, multimerization of antibody fragments can increase molecular weight far above the renal cut-off. AFM13, a tetravalent bispecific anti-CD30 x anti-CD16A TandAb (tandem diabody), with a molecular weight of around 100 kDa, is in phase 2 trials for the treatment of peripheral T cell lymphoma [17]. Bispecific anti-CD137 x anti-EGFR (epidermal growth factor receptor) trimerbodies, with a molecular weight of 160 kDa, have been able to eradicate established tumors in several animal models [18,19]. Therefore, it seems that half-life extension and multimerization strategies would provide Fc-free antibodies with potential for therapeutic opportunities. However, these next-generation antibody therapeutics have faced major bottlenecks in their development, probably associated with challenges in manufacturing, formulation, and stability of clinical grade products. Canonical mAbs are routinely produced in Chinese hamster ovary (CHO) cells, and the majority of purification processes involve protein A-based chromatography, which results in a high degree of purity and recovery in a single step [20]. These procedures provide yields of grams per liter per hour with full-length IgG-based antibodies, which are difficult to achieve with antibody fragments. Adapting proven antibody purification processes to antibody fragments requires different affinity chromatography methods, such as protein L, although a protein L-based production platform may not accommodate all antibody fragments [21].

## 3. Antibody Gene Therapy: A Critical Step toward Clinical Application

An alternative proposed two decades ago was the use of gene-based strategies for in vivo production of antibody fragments [22,23,24]. In 2004, we published a review entitled “Antibodies and gene therapy: teaching old ‘magic bullets’ new tricks” [25] where, we claimed that in vivo production of antibody fragments could result in effective and persistent serum levels compensating for their rapid blood clearance. Furthermore, in vivo secretion could circumvent some of the challenges associated with large-scale production of recombinant antibodies and potentially allow for the design and generation of more complex antibody molecules that may exhibit an improved therapeutic index [26,27,28]. One of the most promising cancer immunotherapy strategies is based on redirecting the activity of T cells using bispecific antibodies, enabling the bridging of cytotoxicity-triggering receptors with a selected tumor-associated antigen (TAA) [26]. TAA-specific T cell-engaging (TCE) bispecific antibodies have revealed outstanding clinical results in some hematological cancers [29]. However, Fc-free BiTEs have a short serum half-life compared with full-length IgG antibodies, and continuous intravenous administration via infusion pumps is necessary to achieve therapeutic serum levels [30]. In 2003, our group reported for the first time a cancer immunotherapy strategy based on the in vivo secretion of a TCE-bispecific antibody [24]. We demonstrated that human cells could be engineered to secrete a functionally active anti-CEA x anti-CD3 Fc-free diabody, which redirects primary T cell cytotoxicity in a CEA-dependent manner and delays the growth rate of human colorectal cancer xenografts in mice [24]. Since then, we have validated this bispecific antibody-based gene therapy strategy using different formats of bispecific antibodies (diabody and BiTE), various types of cell carriers (human T cells, human mesenchymal and hematopoietic stem cells, and human endothelial cells), several gene transfer systems (plasmid DNA and lentiviral vectors), and several mouse cancer models [31,32,33,34,35,36]. Recently, other groups have demonstrated encouraging therapeutic effects in preclinical models of cancer using TCE-secreting T cells engineered with viral vectors [37,38,39], as well as CAR-T cells produced by mRNA technology [40].

## 4. Engineered mRNA: The Last Frontier

In recent years, mRNA-based gene delivery has significantly increased in different therapeutic areas because of the advantages that it possesses over traditional gene delivery platforms (e.g., viral vectors and plasmid DNA) [41,42]. Synthetic mRNA is a short-lived genetic carrier and does not need to enter the cell nucleus for efficient, rapid, and transient protein expression, even in nondividing cells, so there is no risk of insertion into the host genome [43], and it is not hampered by any pre-existing immunity as in the case of viral vectors [44]. However, mRNA can induce Toll-like receptor (TLR)-dependent activation of the innate immune system, thereby affecting the utility of unmodified mRNA-based therapeutics in vivo. Exogeneous mRNA can be sensed by TLRs in the endosomes as well as by receptors such as RIG-I and MDA5, in the cytosol. While improved immune activation can be of interest in vaccination strategies, innate immune sensing can also create an unfavorable environment for the translation of mRNA and thus limit protein expression [45]. The incorporation of modified nucleosides, such as pseudouridine and 5-methylcytidine, in engineered mRNA prevents immune activation and increases the stability and translation of mRNA [46,47]. These chemical modifications, together with efficient delivery methods, such as the use of lipid nanoparticles, have enormously simplified and expanded the use of mRNA for therapeutic applications [48]. One study reported that intravenous administration of engineered mRNA encoding bispecific anti-TAA x anti-CD3 BiTEs induced sustained secretion of functional antibodies, which eliminated established human tumors in mice as effectively as systemically administered purified BiTEs [49]. Despite the interest in mRNA research for therapeutic applications, only a few pre-clinical studies have been reported for mRNA-encoded antibody fragments (Table 3); for full length antibodies, see Deal CE et al. [48]. The first phase 1 clinical trial of an mRNA-encoded antibody [50] has recently been completed, and the results demonstrate the safety of this approach and the potential to reach appropriate levels of circulating antibodies [51]. 

The use of mRNA-based gene delivery platforms will be critical to the widespread clinical application of short-lived Fc-free next-generation antibodies: multivalent and multispecific molecules with a format adapted to the pathological context [5,52]. Unlike systemic administration of purified antibodies, mRNA-based delivery may result in expression for several days and sustained plasma levels, which may allow less frequent dosing when prolonged treatments are required. In addition, the use of synthetic mRNA will allow for the avoidance of manufacturing problems associated with recombinant antibody fragments and next-generation multispecific antibodies, and it is more cost-effective, faster, and flexible to produce. However, for the use of mRNA-encoded antibodies as a competitive therapeutic product and a viable alternative to purified antibodies, further optimization steps are necessary, such as improving formulation technology to reach effective antibody levels and enhanced tolerability, as well as the use of other routes of administration (e.g., intramuscular and subcutaneous) for improved convenience and reduced therapy costs. In addition, targeted delivery of the mRNA to the tissue/organ of interest (e.g., intranasal, intratracheal, or intratumoral) could decrease potential systemic toxicity and reduce the amount of mRNA required to reach therapeutic levels.

## Figures and Tables

**Table 1 antibodies-10-00037-t001:** Fc-free antibody-based therapeutics approved or in review (EU/US).

Non-Proprietary Name	Target	Format	Indication First Approved	First Approval Year (EU/US)
Abciximab	GPIIb/IIIa	Chimeric Fab	Prevention of blood clots in angioplasty	1995 */1994
Ranibizumab	VEGF	Humanized Fab	Macular degeneration	2007/2006
Certolizumab pegol	TNF	Humanized Fab, pegylated	Crohn disease	2009/2008
Blinatumomab	CD19 x CD3	Murine bispecific tandem scFv	Acute lymphoblastic leukemia	2015/2014
Idarucizumab	Dabigatran	Humanized Fab	Reversal of dabigatran-induced anticoagulation	2015/2015
Moxetumomab pasudotox	CD22	Murine scFv + PE immunotoxin	Hairy cell leukemia	2021/2018
Caplacizumab	vWF	Humanized bivalent V_HH_	Acquired thrombotic thrombocytopenic purpura	2018/2019
Brolucizumab	VEGF-A	Humanized scFv	Macular degeneration	2020/2019
Oportuzumab monatox	EpCAM	Humanized scFv + PE immunotoxin	Bladder cancer	NA/In review
Tebentafusp	gp100, CD3	Soluble TCR + scFv ImmTAC	Uveal melanoma	In review/In review

Abbreviations: EpCAM, epithelial cell adhesion molecule; Fab, antigen-binding fragment; GPIIb/IIIa, glycoprotein IIb/IIIa; ImmTAC: immune-mobilizing monoclonal TCR against cancer; PE, Pseudomonas exotoxin A; scFv, single-chain variable fragment; TCR, T cell receptor; TNF, tumor necrosis factor; VEGF-A, vascular endothelial growth factor-A; vWF, von Willebrand factor; *, country-specific approval. Adapted from The Antibody Society. Therapeutic monoclonal antibodies approved or in review in the EU or US. Available online: https://www.antibodysociety.org/resources/approved-antibodies (accessed on 23 September 2021).

**Table 2 antibodies-10-00037-t002:** Fc-free antibody-based therapeutics in late-stage clinical trials.

Company	INN or Code Name	Molecular Format	Target	Most Advanced Phase	Indications
MacroGenics	Flotetuzumab (MGD006)	Humanized bispecific DART	CD123, CD3	Phase 2 (pivotal)	Acute myeloid leukemia (NCT02152956, NCT04582864)
Affimed N.V.	AFM13	Human-bispecific T TandAb	CD30, CD16A	Phase 2 (pivotal)	Peripheral T Cell lymphoma (NCT04101331)
Philogen SpA	Onfekafusp alfa, (L19IL2 + L19TNF)	scFv-based immunocytokines	Fibronectin EDB domain	Phase 3	Melanoma (NCT03567889)
PhaseBio Pharmaceuticals	Bentracimab (PB2452)	Human Fab	Ticagrelor	Phase 3	Reversal of the antiplatelet effects of ticagrelor (NCT04286438)
Taisho Pharmaceutical	Ozoralizumab	Humanized bispecific nanobody	TNF, albumin	Phase 3	Rheumatoid arthritis (JapicCTI-184031, NCT04077567)

Abbreviations: DART, dual-affinity re-targeting antibody; EDB, extradomain-B (fibronectin); Fab, antigen-binding fragment; IL2, interleukin-2; scFv, single-chain Fv; TandAb, tandem diabody; TNF, tumor necrosis factor. Adapted from The Antibody Society. Monoclonal antibodies in late-stage clinical studies. Available online: https://www.antibodysociety.org/members-only/late-stage-clinical-pipeline/ (accessed on 23 September 2021).

**Table 3 antibodies-10-00037-t003:** mRNA-encoded antibody fragments in preclinical studies.

Antibody	Target	Format	Model	Outcome	Ref
RiboMABs	CD3 × CLDN6 CLDN18.2 × CD3 EpCAM × CD3 CD3 × (CLDN6)_2_	Bs (scFv)_2_ Fab-(scFv)_2_	Human PBMC-engrafted NSG mice bearing s.c. human ovarian cancer xenografts	Eradication of tumors (200–300 mm^3^) after one. i.v. infusion a week for 3 weeks	[49]
VNA-BoNTA	Botulism neurotoxin A	Bp (V_HH_)_2_ + ABP	CD1 mice receiving a lethal dose of BoNTA	Survival when treated up to 6 h post-intoxication with i.v. VNA-BoNTA	[52]
VNA-Stx2	*E. coli* Shiga toxin	Bs (V_HH_)_2_ + ABP	CD1 mice receiving a lethal dose of BoNTA	Negative control. No surviving mice in this group	[52]
RSV aVHH	RSV F protein	V_HH_ + GPI anchor	BALB/c mice inoculated i.n. with RSV 1 day post-treatment	RSV titers significantly lower after i.t. aerosol mRNA administration	[53]
RiboBiFE	Mouse FcγRIV x influenza A M2e	Bs (V_HH_)_2_	C57BL/6 mice challenged with i.n. lethal dose of influenza virus 4 h post-treatment	100% survival in wild-type mice receiving i.t. mRNA, 0% in mice FcγRIV^-/-^	[54]

Abbreviations: ABP, albumin-binding peptide; BoNTA, botulism neurotoxin A; Bp, biparatopic; Bs, bispecific; CLDN, claudin; EpCAM, epithelial cell adhesion molecule; Fab, antigen-binding fragment GPI, glycosylphosphatidylinositol; i.n., intranasal; i.t., intratracheal; i.v., intravenous; PBMC, peripheral blood mononuclear cell; RiboBiFE, mRNA-encoded bispecific Fc-receptor engaging; RiboMAB, mRNA-encoded antibody; RSV, respiratory syncytial virus; M2e, matrix protein 2 ectodomain; VNA, VHH-based neutralizing agent.

## Data Availability

Not applicable.

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
