# Peer review of "Engineered mRNA and the Rise of Next-Generation Antibodies"

_2073-4468, 2021, doi:10.3390/antib10040037_

Round 1

Reviewer 1 Report

Review of Antibodies, L. Sanz and L. Alvarez-Vallina (Antibodies (ISSN 2073-4468)

A well-written perspective which provides readers with a concise yet informative overview of advances in the development of Fc-free antibody-based immunotherapeutics for treating cancer. The authors also touch on clinically approved immunotherapeutics in this regard, which is well received.

While there’s no need for major changes, the manuscript will benefit from some minor editing and revision prior to publication:

Line 44: rather 'introduce' than 'inoculate' with regard to injecting antibodies into tissue/disease sites

Table 1: Last row to be formatted

Table 2: comma (,) in first row appearing after ‘MacroGenics’ to be removed

Table 2: formatting required for last column

Lines 55 & 61: Add Fragment crystallizable (Fc)

Line 56 & 62: Add variable fragment (Fv) in scFv abbreviation

Line 65: 'the' is not necessary before 'Holliger and Hudson´s prediction'

Lines 67 - 69: (restructured sentence) Several studies have shown that it is possible to extend the half-life of antibody fragments through fusion with albumin-binding peptides (8) or proteins (9-11).

Line 85: ‘EGFR’

Line 87: 'established'

Line 92: ‘Chinese hamster ovary (CHO) cells…’

Line 93: ‘involves’

Line 101: (restructured sentence part) An alternative proposed two decades ago...

Line 108: '...an improved therapeutic index (26-28)'.

Line 109: 'strategies'

Line 111: 'T cell-engaging'

Line 119: 'delay'

Line 125: '...using TCE-secreting T cells engineered with viral vectors (37-39), as well as CAR-T cells produced by mRNA technology (40)'.

Lines 135-136: (restructured sentence part)'...immune system, thereby affecting the utility of unmodified mRNA-based therapeutics in vivo'.

Line 149: '...single-domain antibodies,...'

Line 163: Authors to kindly elaborate on the following statement: ‘the use of more convenient routes of administration.’

Author Response

Response: We appreciate the reviewer's positive comments as well as the in-depth proofreading. In the revised version we have corrected all the issues pointed out by the reviewer, which are highlighted in red.

Reviewer 2 Report

This is a well written, timely, and insightful review describing the state-of-the art in in vivo production of multispecific antibodies.  I have no major criticisms or concerns but I would suggest to further discuss the drawbacks associated to mRNA-encoded antibodies approach as well as how they can be overcome.

Indeed, an important key point of efficient in vivo production of antibodies by gene transfer approaches is the transduced cell-type, as this might affect the production of circulating molecules at therapeutic levels (both in quantity and long-term production). This needs to be taken into account to improve expression levels and routes of administration and would deserve some further discussion.

Author Response

Response: in the revised manuscript we have modified the final paragraph where we point out the main drawback that need to be addressed in order to make mRNA a competitive therapeutic product and a viable alternative.

Reviewer 3 Report

1. the authors should be insert in the section 4 "Engineered mRNA: the last frontier?" a table with mRNA encoded antibody in clinical trial and/or preclinical trial.

2.  the authors should be insert in the section 4 "Engineered mRNA: the last frontier?" some examples of preclinical studies of mRNA encoded antibody fragment, if exist.

Author Response

Response: According the reviewer recommendation in the revised manuscript we  have included a new Table (Table 3. mRNA-encoded antibody fragments in preclinical studies) and several additional references